# ACE2 localizes to the respiratory cilia and is not increased by ACE inhibitors or ARBs

Ivan T. Lee 🄳 et al.[#]

The coronavirus SARS-CoV-2 is the causative agent of the ongoing severe acute respiratory disease pandemic COVID-19. Tissue and cellular tropism is one key to understanding the pathogenesis of SARS-CoV-2. We investigate the expression and subcellular localization of the SARS-CoV-2 receptor, angiotensin-converting enzyme 2 (ACE2), within the upper (nasal) and lower (pulmonary) respiratory tracts of human donors using a diverse panel of banked tissues. Here, we report our discovery that the ACE2 receptor protein robustly localizes within the motile cilia of airway epithelial cells, which likely represents the initial or early subcellular site of SARS-CoV-2 viral entry during host respiratory transmission. We further determine whether ciliary ACE2 expression in the upper airway is influenced by patient demographics, clinical characteristics, comorbidities, or medication use, and show the first mechanistic evidence that the use of angiotensin-converting enzyme inhibitors (ACEI) or angiotensin II receptor blockers (ARBs) does not increase susceptibility to SARS-CoV-2 infection through enhancing the expression of ciliary ACE2 receptor. These findings are crucial to our understanding of the transmission of SARS-CoV-2 for prevention and control of this virulent pathogen.

[#]A list of authors and their affiliations appears at the end of the paper.

Coronavirus disease 2019 (COVID-19) is an ongoing pandemic infection caused by the positive-sense RNA virus, severe acute respiratory syndrome coronavirus 2 (SARS-CoV-2)[1]. The high transmissibility of the virus, along with case fatality estimates ranging from 1% to above 5%, has generated worldwide concern. Patients with comorbid conditions including hypertension, diabetes, and pulmonary disease are highly represented among hospitalized patients with COVID-19 disease, suggesting the presence of risk factors that may predispose heightened susceptibility to SARS-CoV-2 infection[2–5].

A molecular connection between SARS-CoV-2 and hypertension, in particular, is suggested by the discovery that angiotensin-converting enzyme 2 (ACE2) is the essential receptor for SARS-CoV-2 (refs [6,7]). ACE2 plays an important role in the renin-angiotensin-aldosterone system (RAAS), which consists of a cascade of vasoactive peptides that maintain blood pressure and electrolyte homeostasis. ACE2 converts vasoconstrictor peptides, angiotensin (Ang) II and Ang I, into the vasodilator peptides, Ang (1–7) and Ang (1–9), respectively[8]. These actions counterbalance the enzymatic effect of the related angiotensin-converting enzyme (ACE), which generates Ang II from Ang I.

Angiotensin-converting enzyme inhibitors (ACEI) and angiotensin II receptor blockers (ARBs) are commonly used antihypertensive medications that target components of the RAAS. Several recent correspondences have raised concerns that ACEI and ARBs may increase expression of ACE2 and thereby elevate the risk of infection by SARS-CoV-2, providing a potential explanation for why hypertension is a common comorbidity in patients with COVID-19 (refs [9–12]). This hypothesis is also rooted in human and rodent studies showing upregulation of ACE2 mRNA in the heart, kidney, and urine after ACEI/ARB administration[13–15]. Notably, the effects of ACEI and ARBs on the expression of ACE2 in the respiratory tract have not been previously elucidated. Given the causal role of SARS-CoV-2 in respiratory infections, whether ACE2 expression is altered within the airway of patients taking ACEI or ARBs is a critical question that needs to be addressed to support continued clinical use of these antihypertensive drugs in vulnerable populations.

In this study, we begin by mapping the localization of the SARS-CoV-2 receptor, ACE2, to gain insight into cell tropism and host-viral interactions of SARS-CoV-2. We discover that the ACE2 protein is abundantly expressed in multiciliated airway epithelial cells, spanning from the nasal cavity down to the lower bronchus. Furthermore, we unexpectedly observe robust localization of ACE2 in the motile cilia, a critical structure for mobilizing viral clearance from the airway. We also show evidence for SARS-CoV-2 infection of ciliated cells within the respiratory tract of patients who succumbed to COVID-19. We then apply our finding regarding ACE2 ciliary localization to explore whether ACE2 protein expression is influenced by patient demographics, clinical characteristics, comorbidities, or medication use. We provide mechanism-based evidence that the use of ACEI or ARBs does not increase susceptibility to SARS-CoV-2 infection via its ciliary ACE2 receptors.

## Results

**ACE2 is expressed in the human respiratory tract.** Gene expression analyses have identified ACE2 expression in the nasopharynx, lungs, intestines, kidney, and testis[16], and protein expression studies have largely been concordant with these tissue-specific findings[17,18]. However, a recent manuscript suggested limited to no ACE2 protein expression in the lung, bronchus, and nasopharynx[19]. To understand the precise nature of ACE2 protein expression in tissues relevant for COVID-19, we performed immunohistochemistry using a panel of ACE2 antibodies on human tissue microarrays (TMAs). Consistent with prior studies, we found that several ACE2 antibodies appropriately stain ACE2 in the kidney, testis, seminal vesicles, and intestinal villi (Fig. 1). However, only two antibodies tested (Abcam ab15348 and Sigma HPA000288), stain ACE2 in the CD31$^+$ vascular endothelium (Fig. 2a), where ACE2 expression has also been reported[17]. In the lungs, the anti-ACE2 clone (Abcam ab15348) yielded robust staining of pneumocytes, while the other clones showed negligible or less specific membrane staining (Figs. 1, 2b). After careful antibody titration, clone selection, and validation across multiple tissue types, we report that the overall intensity of ACE2 expression in the lung is low compared to the kidney, testis, and intestinal villi (Supplementary Table 1).

Next, we performed double immunofluorescent staining of ACE2 with mucin 1 (MUC1), an established type II pneumocyte marker, and confirmed that Abcam ab15348 had the most specific staining patterns. Hence this antibody was used in all subsequent experiments. By virtue of co-localization with MUC1$^+$ cells, we definitively demonstrate that the ACE2 protein is expressed within type II pneumocytes of the human lung (Fig. 2b). These data support past findings of ACE2 in presumed type II pneumocytes by single antibody chromogenic staining[17,18] and single-cell RNA-sequencing (scRNA-seq) data showing ACE2 enrichment within type II pneumocytes[20,21]. Overall, our results affirm the localization of the ACE2 protein within the human airway and support the specificity of select commercially available antibodies by orthogonal validation. These antibody testing results also likely serve as a useful resource to guide future protein-based studies.

**ACE2 is expressed in the motile cilia of the airway.** We next investigated the expression of ACE2 protein within the epithelium of the human upper and lower respiratory tract. Recent studies using scRNA-seq have identified ACE2 mRNA expression within ciliated epithelial cells and goblet cells of the nasal cavity[20–22], which together comprise the majority of all differentiated cell types in the airway epithelium. We performed double immunofluorescence staining using anti-ACE2 and anti-acetylated α-tubulin (ACTUB), a marker of the cilia organelle, and discovered that not only is ACE2 expressed in ciliated epithelial cells, but it is robustly expressed in the motile cilia of epithelial cells lining the human nasal turbinate, ethmoid sinus, uncinate process (sinus), trachea, and bronchus (Fig. 3a) compared to appropriate isotype controls (Supplementary Fig. 1). Importantly, since the motile cilium itself does not express mRNA, RNA-based methods would have failed to detect ACE2 expression in the cilia. Next, we examined the mouse respiratory tract and found that ACE2 was similarly expressed within the cilia of the mouse trachea and nasal turbinate (Fig. 3b). Staining of ACE2 within IMCD3 cells, a ciliated kidney epithelial cell line, further confirms ACE2 localization within cilia (Fig. 3c). Finally, overexpression of human ACE2 in mouse IMCD3 cells resulted in a predicted increase in the percentage of ACE2 staining in the primary cilia (Fig. 3d, e), providing compelling evidence of ACE2 localization to the cilia and further validation of antibody specificity. Since both the transcript and protein for transmembrane serine protease 2 (TMPRSS2), a serine protease required for spike protein activation, are also expressed in the motile cilia organelle of the human respiratory tract[18,20,23], we next determined whether SARS-CoV-2 is present within infected ciliated respiratory cells. We performed in situ hybridization (ISH) using a SARS-CoV-2 probe in combination with antibodies to ACE2 and cytokeratin 8 (KRT8), a marker of differentiated epithelial cells, on SARS-CoV-2-infected human sinonasal tissue collected post-mortem at autopsy. SARS-CoV-2 Spike transcripts were detected

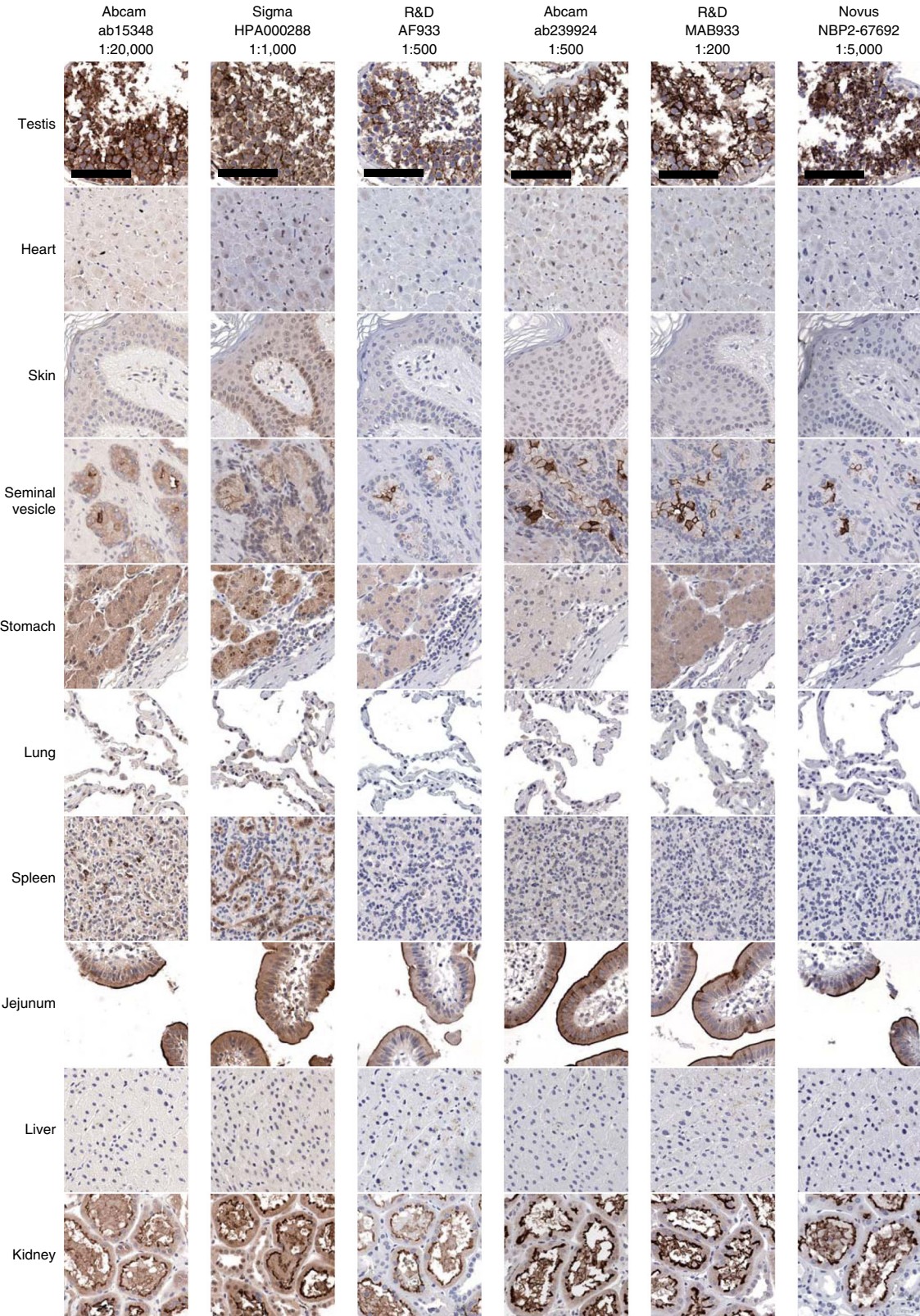

**Fig. 1 Immunohistochemical analysis of ACE2 protein localization across human tissues using multiple anti-ACE2 antibodies.** Representative images of human tissues on a tissue microarray (TMA) stained by chromogenic immunohistochemistry using antibodies targeting the ACE2 protein (brown) and counterstained with hematoxylin (blue). Highest ACE2 expression was observed in the villi of the intestinal tract (jejunum), renal tubules, testis, and glandular cells in the seminal vesicle. Minimal to no/non-specific staining can be seen in the heart, stomach, spleen, skin, and liver. Staining of lung pneumocytes was observed using Abcam ab15348, and less specifically with Sigma HPA000288 (Fig. 2b; Supplementary Table 1). Scale bars: 100 μm.

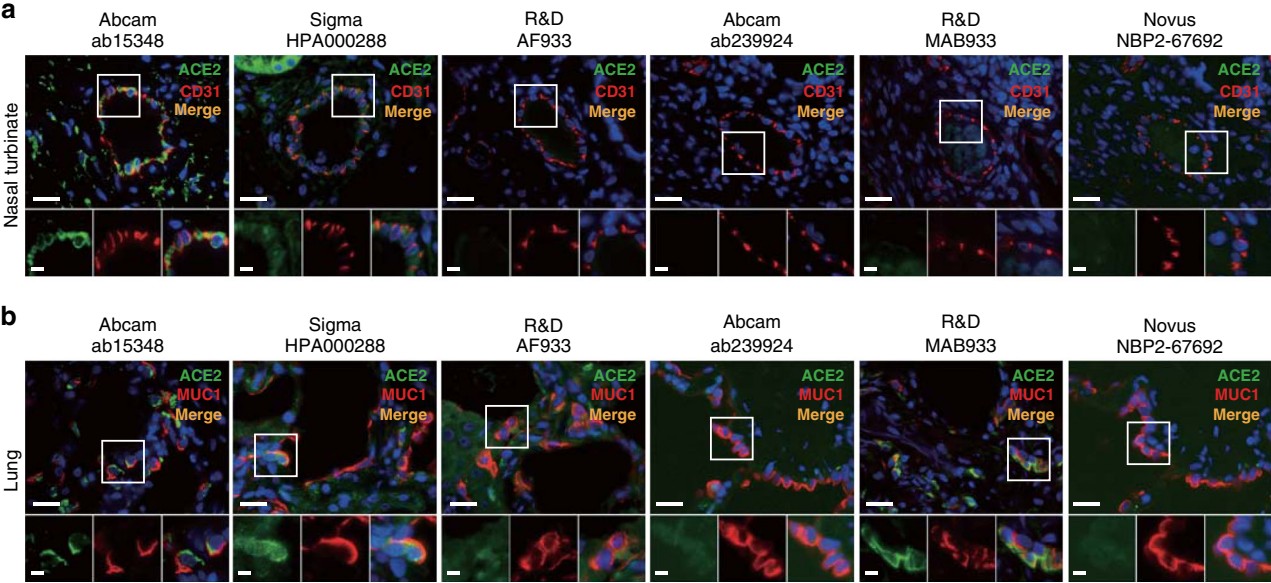

**Fig. 2 ACE2 protein expression within human vascular endothelial cells and the lung. a** Representative double immunofluorescence staining of ACE2 and endothelial cell marker CD31 in the blood vessels of human nasal turbinates using six different anti-ACE2 antibodies and anti-CD31. **b** Double immunofluorescence staining of ACE2 and type II pneumocyte marker mucin 1 (MUC1) in the human lung using six different anti-ACE2 antibodies and anti-MUC1. Abcam ab15348 clone yielded the most robust staining of pneumocytes, while the other clones showed negligible or less specific membrane staining. Scale bars: 20 μm (top) and 5 μm (bottom).

within ciliated epithelial cells expressing ACE2 in the motile cilia (Fig. 3f). Given that motile cilia comprise the outer apical surface of ciliated epithelial cells, this result supports a model in which SARS-CoV-2 first binds ACE2 present in the motile cilia of the upper airway prior to entry into ciliated epithelial cells. Taken together, our data indicate that the respiratory tract motile cilia contain the necessary molecular components to enable cellular entry of SARS-CoV-2.

**Secretory goblet cells of the airway lack ACE2 expression.** Next, given recent scRNA-seq data also reported *ACE2* expression in goblet cells from nasal turbinates and ethmoid tissues of healthy donors and patients with chronic rhinosinusitis (CRS)[20,21], we performed double immunofluorescent staining of ACE2 with Mucin 5AC (MUC5AC), a reliable goblet cell marker. Curiously, the ACE2 protein was not expressed within the secretory goblet cells of the nasal turbinate, uncinate process (sinus), or bronchus (Fig. 4a). Given this disparate result from published single-cell transcript profiling findings, we subsequently performed ISH using an *ACE2* probe in combination with an anti-MUC5AC antibody, and similarly did not detect *ACE2* mRNA expression within goblet cells of the nasal turbinate, uncinate, or trachea (Fig. 4b). These results highlight limitations in the functional interpretation of current single-cell transcriptomic studies, as well as the importance of using targeted transcript and protein validation methods to complement high-throughput analytic approaches. In summary, we find no evidence of ACE2 protein nor mRNA expression in goblet cells of the respiratory airway, suggesting that in contrast to ciliated epithelial cells, secretory goblet cells of the airway epithelium are unlikely to be directly infected by SARS-CoV-2.

**Ciliary ACE2 expression is unchanged by age, sex, or smoking.** We next identified patient factors that may contribute to changes in the expression of ACE2 in the nasal epithelial cilia, as this may have important clinical implications for susceptibility to SARS-CoV-2 transmission. As breathing occurs primarily through the

nose/upper airway, ACE2 in the nasal cilia would be predicted to readily encounter SARS-CoV-2 during transmission. Higher ACE2 expression is correlated with higher pseudotype SARS-CoV-2 and SARS-CoV viral infectivity, suggesting that increased ACE2 levels may predispose individuals to SARS-CoV-2 infection[24–26]. We leveraged our existing comprehensive human nasal tissue bank, which contains detailed demographics, medical, social, and medication history from patients who have donated their upper airway tissues from three academic medical centers (Stanford University Hospital, National Taiwan University Hospital (NTUH), and China Medical University Hospital (CMUH)) between 2018 and 2020, to characterize whether ACE2 expression in the upper respiratory cilia is affected by specific patient characteristics (Table 1).

We first determined the extent to which ACE2 expression may differ by age, sex, and smoking status — three covariates that have been associated with COVID-19 disease severity[27,28]. Across all three sample cohorts, we identified no significant differences in ACE2 expression based on age (≥65 years), sex, or smoking status (Fig. 5a). These results differ from some, but not all, recent gene expression studies comparing ACE2 expression in patients with varying demographics and smoking status[29–31]. Our results suggest that host factors outside of ACE2 expression may determine why males, patients of older age, and smokers are epidemiologically linked to COVID-19 susceptibility.

**Ciliary ACE2 is unchanged in CRS across the sinonasal airway.** We next examined whether ACE2 expression in the upper respiratory cilia differs between healthy donors versus patients with chronic rhinosinusitis (CRS), a non-malignant chronic inflammatory disease of the paranasal sinuses that presents either with benign nasal polyps (CRSwNP) or without nasal polyps (CRSsNP). Across all three patient cohorts, no significant differences were noted between healthy donors and patients with chronic rhinosinusitis with or without nasal polyps (Fig. 5b). There were also no observed differences in ACE2 expression between anatomical regions within the sinonasal cavity (Fig. 5c). These results suggest that patients with CRS may not be at a

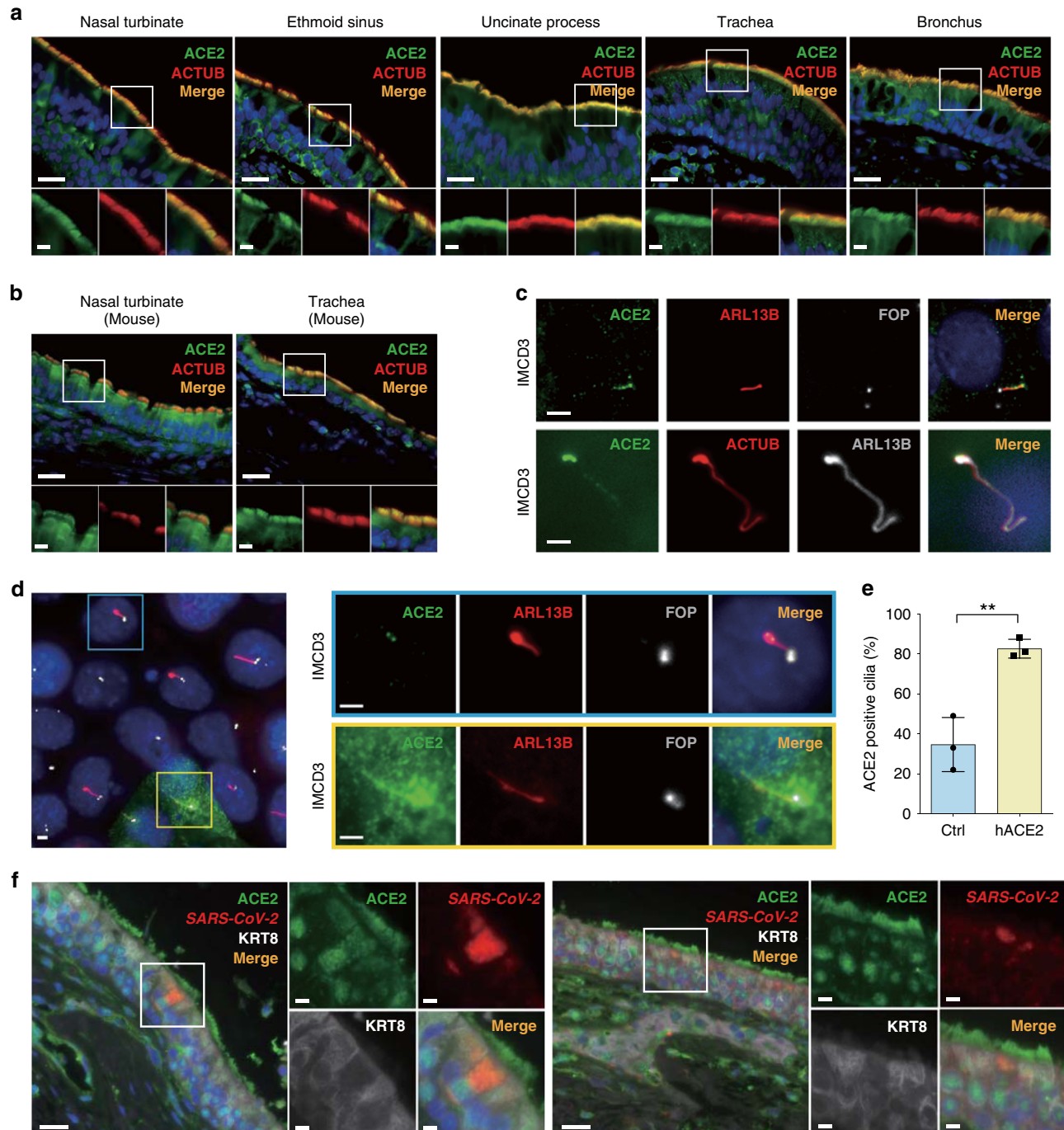

**Fig. 3 ACE2 protein expression in the cilia organelle of upper and lower respiratory tract epithelia and in a ciliated kidney epithelial cell line. a** Representative double immunofluorescence staining of ACE2 and acetylated α-tubulin (ACTUB) on normal human nasal turbinate, ethmoid sinus, uncinate process (sinus), trachea, and bronchus, using anti-ACE2 and anti-ACTUB antibodies, respectively. **b** Representative double immunofluorescence staining of ACE2 and ACTUB on normal C57BL/6J mouse nasal turbinate and trachea. **c** Immunofluorescent staining of (top panel) ACE2, cilia marker ADP-ribosylation factor-like protein 13B (ARL13B), and cilia centrosome marker FGFR1 oncogene partner (FOP); (bottom panel) ACE2, and cilia markers ACTUB and ARL13B in a ciliated mouse cell line, IMCD3. **d** Immunofluorescent staining of ACE2 in the primary cilia of IMCD3 cells transiently transfected with human *ACE2* (yellow outline) compared to endogenous mouse ACE2 (blue outline). **e** Quantified percentages of endogenous ACE2-positive cilia (34.67 ± 13.58%; control (Ctrl)) versus cilia with overexpressed human *ACE2* (82.67 ± 4.73%). Ciliated cells were identified by staining of ARL13B. Error bars represent mean ± SD. ($n = 100$ cells examined per experiment over three independent experiments). (Two-tailed Student's *t* test, **$p = 0.004$). **f** Representative multiplexed images of in situ hybridization against the *SARS-CoV-2 Spike* mRNA, in combination with immunofluorescence staining of ACE2 and the differentiated epithelial cell marker cytokeratin 8 (KRT8). *SARS-CoV-2 Spike* mRNA expression (red) was detected within ciliated epithelial cells containing motile cilia positive for ACE2 (green). The nuclei were stained using DAPI (blue) as a counterstain. Scale bars: 20 µm (**a**, **b** top panels; **f** large panels); 5 µm (**a**, **b** bottom panels; **f** small panels); 2 µm (**c**, **d**).

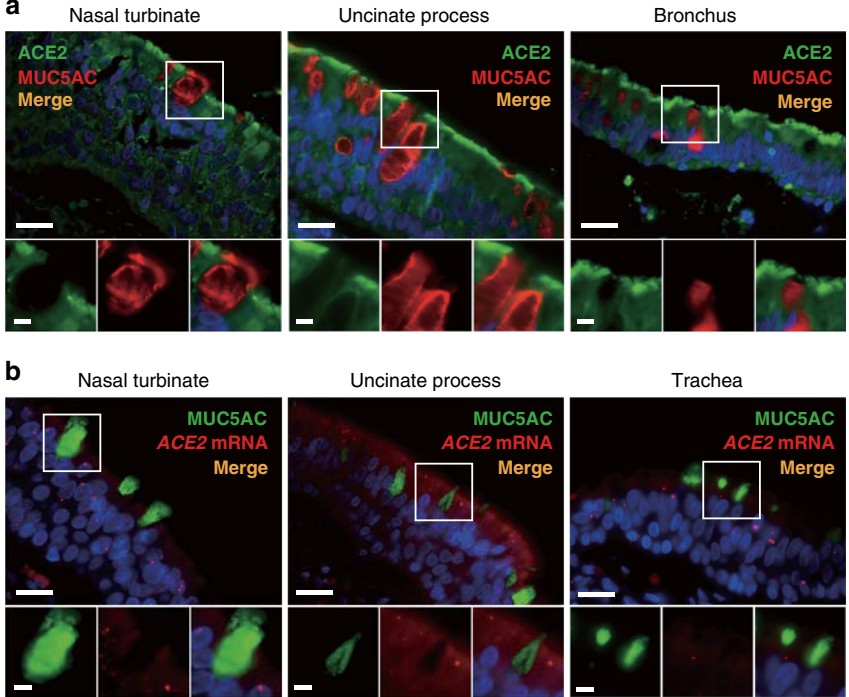

**Fig. 4 ACE2 protein and mRNA expression are not found in secretory goblet cells of the human airway. a** Representative immunofluorescence double staining of ACE2 and mucin 5AC (MUC5AC) reveals absence of co-localization of ACE2 within secretory goblet cells in the human nasal turbinate, uncinate process, and bronchus. **b** Representative in situ hybridization using an *ACE2* probe in combination with an anti-MUC5AC antibody. *ACE2* mRNA expression (red dots) was not detected within goblet cells marked by MUC5AC in the nasal turbinate, uncinate process, and trachea. Nuclei were stained using DAPI. Scale bars: 20 μm (top) and 5 μm (bottom).

higher risk of SARS-CoV-2 infection, although it would be important to assess other factors such as inflammation in future studies. There are currently no epidemiological studies that have assessed the prevalence of COVID-19 in patients with CRS to our knowledge.

**Ciliary ACE2 is not increased in patients taking ACEI/ARBs.** Finally, we identified patients within our sinonasal tissue bank who have been taking either ACEI or ARBs for at least six continuous months prior to sinonasal surgery and compared their ACE2 expression to controls matched for age, sex, and smoking status who have never taken ACEI/ARBs. Baseline characteristics of the patients are not significantly different except for older age in patients taking ARBs compared to the control group in the CMUH cohort (Table 1). We find that ciliary ACE2 expression is slightly, but statistically significantly, decreased in patients taking ACEI compared to matched controls in the Stanford cohort, whereas ACE2 expression was not significantly different in patients taking ARBs compared to controls in all three patient cohorts (Fig. 6a). We were unable to identify patients taking ACEI in the two Taiwanese cohorts, likely because ARBs are strongly preferred over ACEI for management of hypertension in Taiwan[32]. Subgroup analysis comparing ACEI and ARBs treatment groups to only controls with hypertension (on other medications) revealed a similar trend of lower ACE2 expression in the ACEI and ARB group, although statistical significance was not attained likely due to reduced sample size (Fig. 6b). When patients from all three cohorts were combined as a normalized Z-score to increase power, ACE2 expression in the patients taking ACEI was significantly lower compared to controls (Fig. 6c). Further subgroup analysis comparing ACEI and ARBs treatment groups to controls of similar age, sex, or smoking status revealed a similar trend of non-significant but lower ACE2 expression in the

ACEI and ARB group among most cohort groups (Fig. 6d–f). Above all, these results indicate that the use of ACEI or ARBs does not increase ACE2 expression in the upper respiratory cilia, and therefore patients taking ACEI or ARBs are likely at no greater risk of SARS-CoV-2 transmission than individuals not on these medications.

## Discussion

We report several notable discoveries in this basic and translational research study. First, we discover that ACE2 is robustly expressed in the motile cilia of the respiratory tract. This precise subcellular organelle localization has never been previously reported to our knowledge. Cilia are microscopic, finger-like protrusions that project above the apical surfaces of epithelial cells into the nasal and bronchial airway lumen. Acceleration of ciliary beating triggered by irritants, inflammatory signals, and viral pathogens increases mucus flow to sweep foreign substances out of the respiratory tract. This discovery has several important implications. Since ~80% of the human respiratory epithelium from the nasal cavity down to the lower bronchus is densely covered with cilia (50–200 cilia per epithelial cell)[33], the presence of ACE2 in the respiratory cilia represents an exceptionally large surface area for SARS-CoV-2 binding and cell entry (Fig. 7). This may in part explain the high transmissibility of SARS-CoV-2 and clearly supports the use of face masks to decrease upper airway transmission. Furthermore, there are currently no targeted approaches to mitigate COVID-19 by inhibiting ciliary ACE2. Given the ease of local delivery of topical nasal sprays and large-volume sinus irrigations to the sinonasal cavity, we believe that nasally administered therapeutic or prophylactic approaches to block SARS-CoV-2 entry through ciliary ACE2 should be explored expeditiously.

**Table 1 Demographic summary of patients.**

|  |  | Control | ARBs | ACEI | *p* value |
|---|---|---|---|---|---|
| Stanford (*n* = 28) | Number of patients | 16 | 7 | 5 | – |
|  | Age (years) | 53 ± 18 | 68 ± 17 | 66 ± 12 | 0.09 |
|  | Sex | – | – | – | 0.90 |
|  |   Male, *n* (%) | 10 (63) | 5 (71) | 3 (60) | – |
|  |   Female, *n* (%) | 6 (38) | 2 (29) | 2 (40) | – |
|  | Smoking | – | – | – | 0.89 |
|  |   Current, *n* (%) | 0 (0) | 0 (0) | 0 (0) | – |
|  |   Former, *n* (%) | 5 (31) | 2 (29) | 1 (20) | – |
|  |   No, *n* (%) | 11 (69) | 5 (71) | 4 (80) | – |
|  | Hypertension, *n* (%) | 6 (38) | 7 (100) | 5 (100) | <0.01 |
|  | Sinus diseases | – | – | – | 0.27 |
|  |   Control, *n* (%) | 5 (31) | 1 (14) | 0 (0) | – |
|  |   CRSsNP, *n* (%) | 4 (25) | 1 (14) | 3 (60) | – |
|  |   CRSwNP, *n* (%) | 7 (44) | 5 (71) | 2 (40) | – |
| NTUH (*n* = 31) | Number of patients | 19 | 12 | – | – |
|  | Age (years) | 62 ± 11 | 62 ± 12 | – | 0.93 |
|  | Sex | – | – | – | 0.48 |
|  |   Male, *n* (%) | 10 (53) | 8 (67) | – | – |
|  |   Female, *n* (%) | 9 (47) | 4 (33) | – | – |
|  | Smoking | – | – | – | 0.70 |
|  |   Current, *n* (%) | 1 (5) | 0 (0) | – | – |
|  |   Former, *n* (%) | 2 (11) | 1 (8) | – | – |
|  |   No, *n* (%) | 16 (84) | 11 (92) | – | – |
|  | Hypertension, *n* (%) | 10 (53) | 12 (100) | – | <0.01 |
|  | Sinus diseases | – | – | – | 0.23 |
|  |   Control, *n* (%) | 4 (21) | 0 (0) | – | – |
|  |   CRSsNP, *n* (%) | 1 (5) | 1 (8) | – | – |
|  |   CRSwNP, *n* (%) | 14 (74) | 11 (92) | – | – |
| CMUH (*n* = 25) | Number of patients | 17 | 8 | – | – |
|  | Age (years) | 50 ± 10 | 62 ± 6 | – | <0.05 |
|  | Sex | – | – | – | 0.20 |
|  |   Male, *n* (%) | 7 (41) | 6 (75) | – | – |
|  |   Female, *n* (%) | 10 (59) | 2 (25) | – | – |
|  | Smoking | – | – | – | 0.29 |
|  |   Current, *n* (%) | 3 (18) | 3 (38) | – | – |
|  |   Former, *n* (%) | 2 (12) | 2 (25) | – | – |
|  |   No, *n* (%) | 12 (71) | 3 (38) | – | – |
|  | Hypertension, *n* (%) | 3 (18) | 8 (100) | – | < 0.01 |
|  | Sinus diseases | – | – | – | 0.49 |
|  |   Control, *n* (%) | 6 (35) | 1 (13) | — | – |
|  |   CRSsNP, *n* (%) | 5 (29) | 3 (38) | — | – |
|  |   CRSwNP, *n* (%) | 6 (35) | 4 (50) | — | – |

Demographics and characteristics of enrolled patients. For continuous variables, Kruskal–Wallis test was used for three group comparisons and two-tailed Mann–Whitney test was used for two-group comparisons. For nominal data, the two-sided $\chi^2$ test was used.

The endogenous function of ACE2 in the cilia may be distinct from its role in the RAAS and remains to be determined. A recent study suggests a link between ciliary signaling, hypertension, and control of the RAAS system[34], which raises questions as to whether SARS-CoV-2, through interaction with ACE2, might interfere with ciliary function during late viremia. Could viral binding to ciliary ACE2 cause an increase in ciliary beating and changes in the normal dynamics of mucociliary clearance, potentially linked to the dry cough often observed in COVID-19 patients? Could the cilia dysfunction also play a role in kidney failure in patients with severe COVID-19? Studies are underway to further explore these questions.

It is intriguing to speculate how SARS-CoV-2 evolved to target the easily accessible ACE2 receptors in the motile cilia. Nasal epithelial cells have been known to be early sites of viral contact for many respiratory viruses including SARS-CoV[35], MERS-CoV[36], respiratory syncytial virus (RSV)[37], and other viral pathogens[38–40], which each exploit respective host proteins harbored within these luminal epithelial cells. CX3CR1, the receptor for RSV, in particular, is expressed in the motile cilia within respiratory epithelial cells[41], much like our findings of ACE2 expression in the cilia. In two recently published works, although the authors did not comment on this finding, the SARS-CoV-2 antigen was clearly visualized in the motile cilia organelle of SARS-CoV-2-infected cynomolgus macaques and infected bronchial cells ex vivo[42,43]. The related SARS-CoV has also been observed in the cilia of the airway[35]. These results accentuate our discovery of ACE2 in the cilia and collectively, strongly suggest that the motile cilia of the airway are the initial or early subcellular sites of SARS-CoV-2 entry.

A second point of interest in this study is our observation that ACE2 is not expressed in goblet cells of the respiratory tract. Secretory goblet cells make up ~20% of the epithelial cells in the airway[33] and play an important function in mucus production for motile cilia to sweep out unwanted substances during mucociliary clearance. We show on both the mRNA and protein level that ACE2 is not present within goblet cells of the upper and lower respiratory tract. We can only speculate why recent scRNA-seq

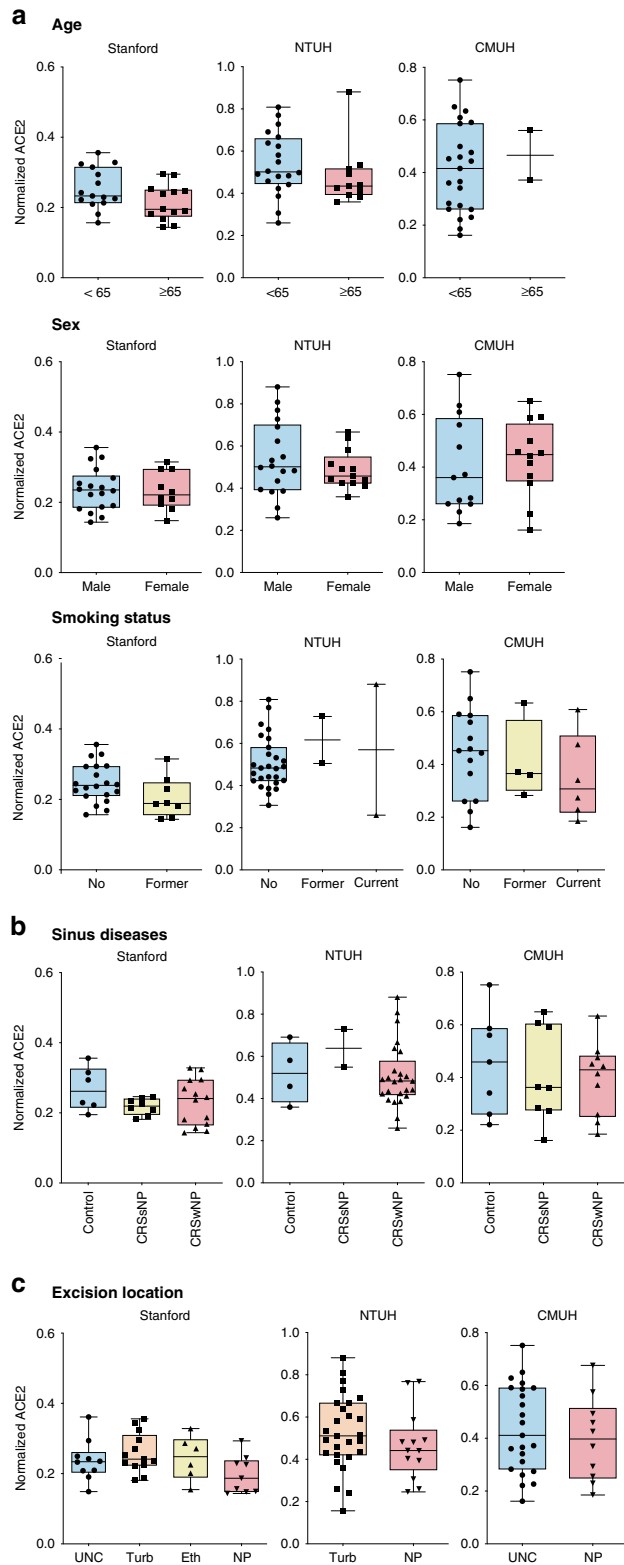

**Fig. 5 Comparison of ciliary ACE2 protein expression by age, sex, smoking status, sinus disease, and anatomical region. a** No statistically significant changes in ACE2 expression was detected among patients less than or greater than 65 years of age, males versus females, and patients with varying smoking history. (Two-tailed Mann–Whitney test or Kruskal–Wallis test, $p > 0.05$). **b** No statistically significant difference in ACE2 expression was observed between healthy controls and patients with chronic rhinosinusitis with polyps (CRSwNP) or without polyps (CRSsNP). (Kruskal–Wallis test, $p > 0.05$). **c** No statistically significant difference in ACE2 expression was noted between distinct human nasal tissue sites/ regions. (Two-tailed Mann–Whitney test or Kruskal–Wallis test, $p > 0.05$). UNC uncinate process, Turb nasal turbinates, Eth ethmoid sinus, NP benign nasal polyps. The bottom and top of the box plots represent the 25th and 75th percentiles, respectively. The bands within the box show the median value, and the whiskers extending from both ends of the boxes are minimum and maximum values. Each dot represents one patient.

clinical characteristics, and medication use. We found no significant differences in ciliary ACE2 expression based on age (≥65 years), sex, or smoking status. It is worth noting that although smoking is associated with shortened cilia length[44], this does not imply that smoking is protective against COVID-19, as shortened cilia most often suggest defects in ciliary signaling, which in turn may hinder mucociliary clearance. On the contrary, our data show that ciliary ACE2 expression is not decreased in smokers (Fig. 5a, lower panel), which is in line with a recent meta-analysis showing that smoking is not a protective factor, but rather a risk factor for progression of COVID-19 (ref. [27]).

Notably in the present study, we found that nasal ciliary ACE2 levels are not increased in patients taking ACEI or ARBs. This conclusion also applies when comparing patients with hypertension taking other classes of antihypertensives to patients taking ACEI or ARBs. The same result is true among both sexes, younger and older persons, and non-smokers taking ACEI/ARBs (Fig. 6). Several observational studies have published findings using patient databases that support the absence of association between ACEI/ARBs use and COVID-19 (refs [45–48]). Our study results concur with these findings, but in addition, provide a mechanistic explanation for why ACEI/ARBs use does not increase the risk of COVID-19; namely that the upper respiratory ciliary ACE2 is not increased in patients taking ACEI/ARBs. This mechanism-based explanation is important as there are limitations to clinical observational studies such as the variable sensitivity/specificity of current COVID-19 testing methods and potential inaccuracies in the electronic health record. Our study has the advantage in that the use or non-use of ACEI/ARBs were verified for every patient through direct interview by research personnel, rather than just by electronic database searches, where true drug consumption in the treatment group and the lack of medication use in controls cannot be ascertained. Lastly, it is interesting that we found decreased ciliary ACE2 expression in patients taking ACEI, suggesting a potential protective role for ACEI in SARS-CoV-2 infection. However, this association needs to be considered with extreme caution as the sample size of our ACEI group is low, and our study is an observational study, and therefore no causal inference can be made.

Given that the SARS-CoV-2-infected nasal tissue shown was sampled during autopsy of a patient in the advanced, late-stages of COVID-19, the staining characteristics seen may contain inherent artifacts of postmortem tissue preparation and may not be representative of early stages of SARS-CoV-2 binding and cell entry. The translational portion of our study also has expected limitations. First, sample sizes in the treatment groups, particularly the ACEI group, were relatively small, which may increase

data have identified *ACE2* mRNA expression within goblet cells. Goblet cells are inherently "sticky" due to mucopolysaccharides and this characteristic may perhaps lead to technical difficulties in the interpretation of expression profiling data. Our results further narrow down the ciliated epithelial cells as a critical cell type targeted by SARS-CoV-2 during viral transmission.

An additional aim of our study focused on assessing changes in nasal ciliary ACE2 among patients of varying demographics,

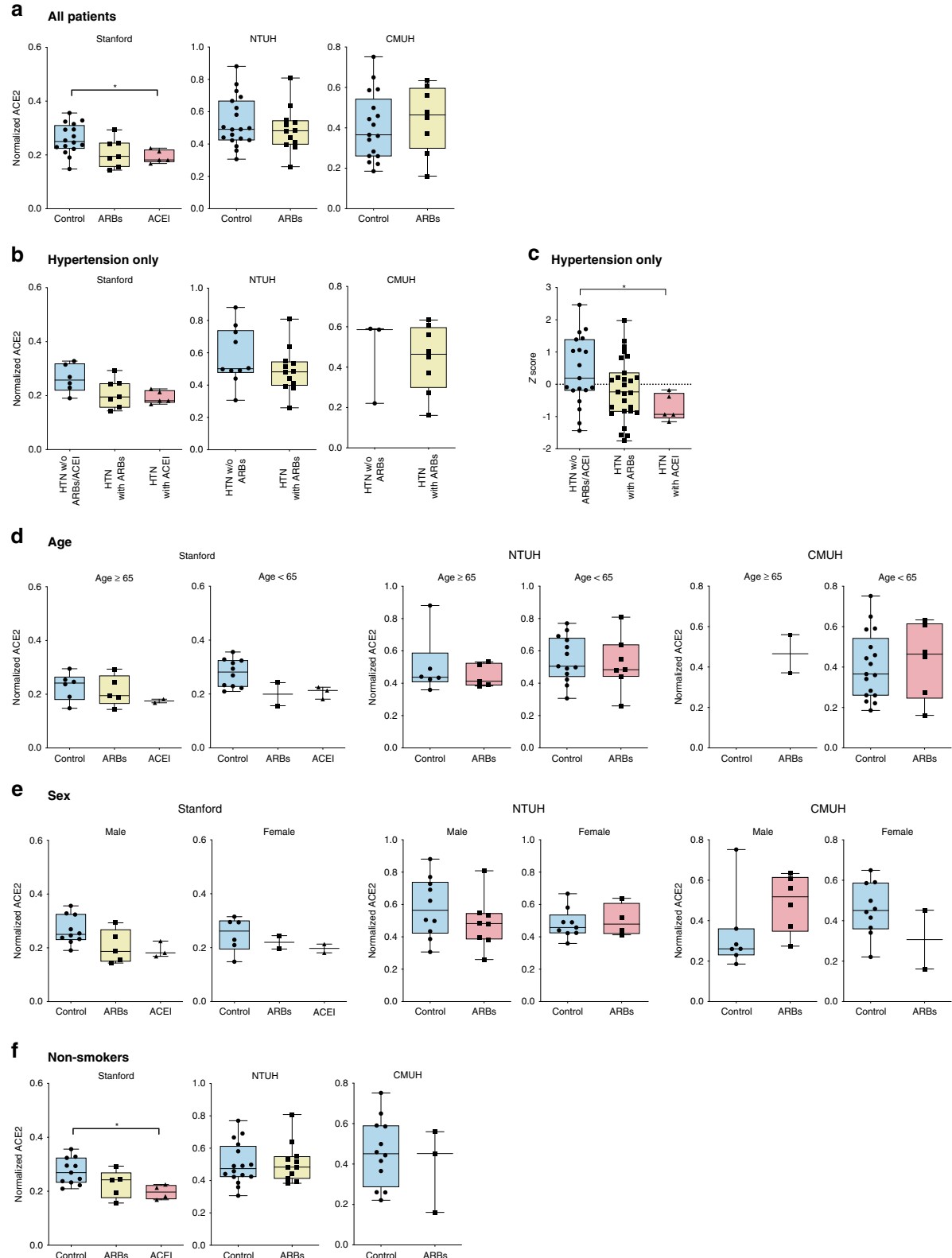

the possibility of a Type II error. However, our hypothesis was that ACEI/ARBs treatment would increase ACE2 expression. Instead, we observe a decrease in ACE2 among the treatment groups (with the ACEI group reaching statistical significance). The true probability that ACE2 is increased following ACEI or ARBs treatment is therefore exceedingly low. Second, recall bias may be present as information on ACEI/ARBs usage was collected retrospectively, although we anticipate this bias to be low as all patients' use/lack of use of long-term ACEI/ARBs were confirmed by direct interview and verified in the electronic medical or pharmacy record. Lastly, similar to all available clinical studies on the effect of ACEI/ARBs in relation to COVID-19, this study was a non-interventional study in which the use of anti-hypertensive agents for each patient was not randomized, and

**Fig. 6 ACE2 expression in the nasal cilia is not increased in patients taking ACEI or ARBs. a** Quantification of ACE2 in controls and patients taking ARBs and ACEI. In the Stanford cohort, ACE2 is slightly but statistically significantly lower in patients taking ACEI (0.19 ± 0.02) compared to controls (0.26 ± 0.06). (Kruskal–Wallis test $p = 0.021$; Dunn's multiple comparison post-hoc test, *adjusted $p = 0.043$). There were no statistically significant differences in ACE2 expression between patients taking ARBs and controls in the Stanford, National Taiwan University Hospital (NTUH), and China Medical University Hospital (CMUH) cohorts. **b** In the Stanford cohort, when including only controls with hypertension (HTN) on other medications ("HTN w/o ARBs/ACEI"), ACE2 expression was statistically different between the groups (Kruskal–Wallis test, $p = 0.044$) but Dunn's multiple comparison post-hoc test did not reveal any statistical significance between the three groups. No statistically significant differences were seen among patients taking ARBs compared to controls. **c** When cohorts from all three institutions were normalized by Z-score and integrated, patients taking ACEI ($-0.72 ± 0.42$) had a lower ACE2 expression compared to controls with hypertension (0.41 ± 1.07). (Kruskal–Wallis test, $p = 0.032$; Dunn's multiple comparison post-hoc test, *adjusted $p = 0.043$). Patients taking ARBs ($-0.15 ± 0.95$) showed a trend towards lower ACE2 compared to controls with hypertension, but this was not statistically significant. **d** ACE2 expression among patients of older (≥65 years) and younger (<65 years) age taking ARBs or ACEI was not statistically divergent from control patients of the same age group. (Kruskal–Wallis test, $p > 0.05$). **e** ACE2 expression among male and female patients on ARBs or ACEI trended comparably or lower than same-sex controls except for males taking ARBs in the CMUH group who showed a trend towards higher ACE2 expression. No statistically significant differences were observed. (Kruskal–Wallis test, $p > 0.05$). **f** Among non-smokers, there was a statistically significant trend towards lower ACE2 expression in patients taking ACEI compared to controls in the Stanford group (Kruskal–Wallis test, $p = 0.021$; Dunn's multiple comparison post-hoc test, *adjusted $p = 0.035$). No statistical significance was observed with the non-smokers on ARBs. All data are noted as mean ± SD. Kruskal–Wallis test was used for three group comparisons and two-tailed Mann–Whitney test was used for two-group comparisons. Box plots are similar in format to Fig. 5.

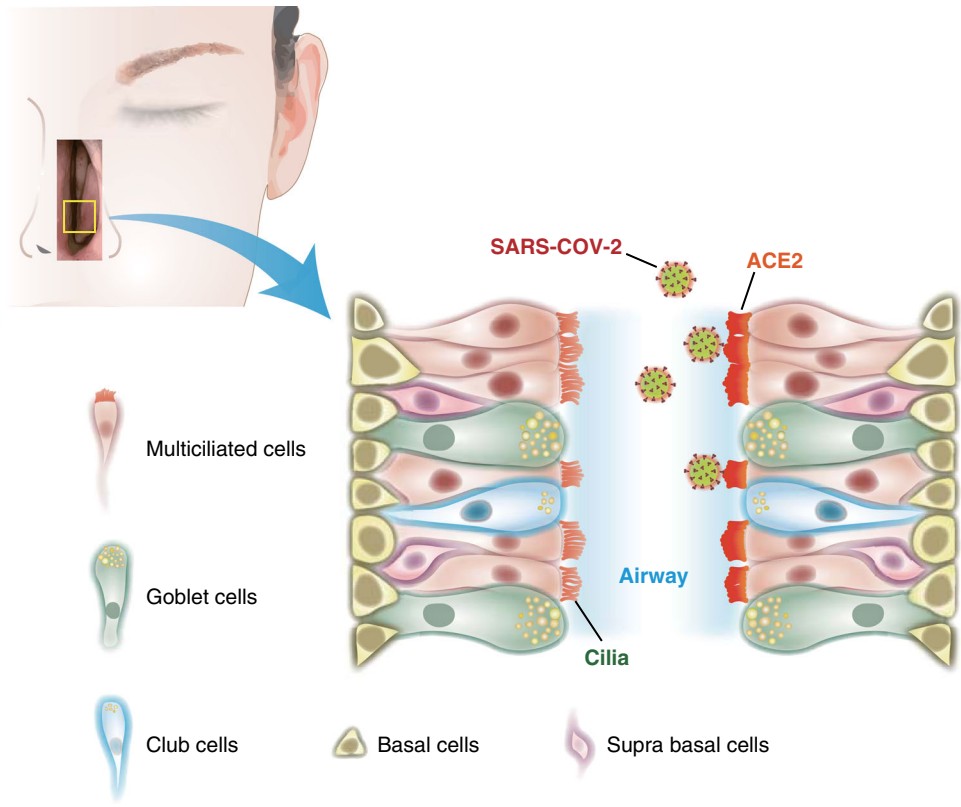

**Fig. 7 Illustration of ACE2 expression in the motile cilia of epithelial cells in the nose, paranasal sinuses, trachea, and lower airways.** The luminal differentiated airway epithelial cells consist of ciliated columnar cells (~80%) and secretory goblet cells (~20%). Club cells are infrequently found in the human upper airway. The basal cell layer, which faces the lamina propria, is comprised of both basal and suprabasal cells, which are considered multipotent progenitors capable of renewing the airway epithelium. This schematic depicts how SARS-CoV-2 may bind to ACE2 expressed on the cilia of the nasal cavity following exposure to respiratory droplets or airborne particles.

therefore the ability to make a causal inference is limited due to potential confounding factors. Although we minimized these elements by using controls matched for age, sex, and smoking status, it is possible that there are other yet unidentified factors for SARS-CoV-2 infection. Randomized placebo-controlled clinical trials (NCT04338009 and NCT04312009) will be necessary and are currently underway to further ascertain the impact of continuation versus discontinuation of ACEI and ARBs on outcomes in patients with COVID-19.

In conclusion, we find that ACE2 protein expression is not only present in epithelial cells lining the human respiratory tract, but that on a subcellular level, it is enriched in the motile cilia of

the respiratory airway. Projecting above the epithelium of the nasal cavity where normal airflow generally occurs, the motile cilia of the airway are anatomically poised to be the initial or early subcellular sites of SARS-CoV-2 viral entry. The identification of ACE2 in the motile cilia will help guide future functional studies and the development of potential targeted therapies that block SARS-CoV-2 viral entry and infection routes via the cilia. Finally, our findings support the conclusion from several professional medical societies that have endorsed the maintenance of standard ACE inhibitors and ARBs therapy during the ongoing SARS-CoV-2 outbreak, although those prior recommendations were provided in the absence of mechanistic evidence that ACE2 levels

may be increased by ACEI/ARBs in airway tissues. Here, we reveal mechanism-based evidence that ACE2 levels in the respiratory tract are indeed not increased in patients taking long-term ACEI and ARBs, suggesting that these medications can be safely continued as standard antihypertensive therapeutics.

## Methods

**Human nasal tissue specimen collection.** Tissues from the nasal cavity and the paranasal sinuses were collected from both healthy control donors and patients with chronic rhinosinusitis from 2018 to 2020 at the Stanford Sinus Center, National Taiwan University Hospital (NTUH), and China Medical University Hospital (CMUH) in Taiwan. Controls represented patients without history, endoscopic, or radiographic evidence of sinus disease, but underwent sinus procedures for surgical access such as for repair of cerebrospinal fluid leaks. Detailed patient characteristics including demographics, medical history, and past medication use were collected in parallel with tissue sample acquisition. Patient data, including medication history, were independently verified through direct interview by a research technician/physician and by a questionnaire additionally administered on the day of surgery to confirm accuracy of existing records derived from patients' electronic medical or pharmacy records. Samples were included if the use or non-use of ACEI or ARBs could be confirmed in-person and by electronic medical or pharmacy records. All tissue specimens and patient medical record information, including demographics, medical/social history, and medications were collected under approved Institutional Review Board (IRB) protocols in accordance with the regulations of the Research Compliance Office at the respective institutions (protocol ID 18981 at Stanford University, protocol ID 201805020RINA at National Taiwan University, and protocol ID CMUH107-REC3-142(CR-1) at China Medical University). All subjects provided informed consent. Following surgical excision, sinonasal specimens were placed in physiologic saline, immediately transported to the lab, and placed in 10% neutral buffered formalin for 24–48 h before paraffin embedding. Nasal turbinate, uncinate process, and ethmoid sinus tissues were placed into EDTA for bone decalcification prior to embedding into tissue blocks. SARS-CoV-2 infected sinonasal tissue was obtained during autopsy and processed as previously described[49], and approved by the ethics commission of Northern Switzerland (EKNZ; study ID: 2020-00969). All COVID-19 patients or their relatives consented to the use of tissue for research purposes.

**Human lung, bronchial, and tracheal tissues.** Formalin-fixed, paraffin-embedded (FFPE) tissue blocks from Stanford Pathology archives were selected based on normal histology using hematoxylin-eosin (H&E) stained tissue sections for evaluation. Normal histology was reconfirmed by a board-certified pathologist (C.M. S.) in the Nolan laboratory.

**Tissue microarrays.** FFPE blocks from 35 normal tissues were retrieved from the tissue archive at the Institute of Pathology, University of Bern, Switzerland. Normal tissue regions were annotated on corresponding H&E-stained sections by a board-certified pathologist (C.M.S.). TMAs with 0.6 mm diameter cores were assembled using a TMA Grand Master automated tissue microarrayer (3DHistech). The use of patient tissue samples was approved by the local Ethics Committee of the Canton of Bern (KEK 200/2014).

**Chromogenic immunohistochemistry.** After deparaffinization and rehydration, slides were blocked for endogenous peroxidase in 3% hydrogen peroxide for 15 min at room temperature. Heat-Induced Epitope Retrieval (HIER) was performed with Dako Target Retrieval Solution, pH 9 (S236784-2, DAKO Agilent) at 95 °C for 25 min. In all, 2.5% horse serum was used for blocking for 30 min at room temperature followed by incubation overnight at 4 °C using one of the following primary antibodies (final titrations in parentheses): rabbit anti-ACE2 (1:20,000; Abcam ab15348), rabbit anti-ACE2 (1:1,000; Sigma HPA000288), goat anti-ACE2 (1:500; R&D Systems AF933), rabbit anti-ACE2 (1:500; Abcam ab239924), mouse anti-ACE2 (1:200; R&D Systems MAB933), and rabbit anti-ACE2 (1:5,000; Novus NBP2-67692). ImmPRESS HRP anti-rabbit, anti-mouse, or anti-goat IgG polymer detection kit (Vector Laboratories, Burlingame, CA) was used as the secondary antibody link for 30 min at room temperature. Five-minute Tris/tween buffer washing steps were performed between each incubation step. The immune complexes were visualized with ImmPACT DAB Peroxidase (HRP) Substrate kit (Vector Laboratories). Cell nuclei were counterstained with hematoxylin. Images were scanned and digitized using the Aperio AT2 whole slide scanner and viewed using Aperio ImageScope (v12.4.3.5008) software.

**IMCD3 cell culture, transfection, and staining.** Mouse IMCD-3 (ATCC CRL-2123™) cells were grown in DMEM (11995065, Thermo Fisher Scientific) supplemented with 10% FBS (100–106, Gemini), GlutaMax supplement (35050-079, Thermo Fisher Scientific), and 100 U/mL Penicillin-Streptomycin (15140163, Thermo Fisher Scientific) at 37 °C in 5% $CO_2$. For transient transfection of IMCD3 cells, IMCD3 cells were grown to ~80% confluence and transfected with 5 µg of full-length *ACE2* plasmid DNA/$1 \times 10^6$ cells using Fugene6 (Promega). Prior to

staining, cells were grown on 12 mm round coverslips and fixed with 4% paraformaldehyde in PBS at room temperature for 10 min. Samples were blocked with 5% normal donkey serum (017-000-121, Jackson ImmunoResearch) in IF buffer (3% BSA and 0.4% saponin in PBS) at room temperature for 30 min before overnight incubation with primary antibodies in IF buffer at 4 °C. Following five washes with IF buffer, samples were then incubated with fluorescent-labeled secondary antibody at room temperature for 1 h, followed by a 5-min incubation with 4′,6-diamidino-2-phenylindole (DAPI) in PBS at room temperature for 5 min and five washes with IF buffer. Coverslips were mounted with Fluoromount-G (0100-01, SouthernBiotech) onto glass slides. Images were acquired on an Everest deconvolution workstation (Intelligent Imaging Innovations) equipped with a Zeiss AxioImager Z1 microscope, CoolSnapHQ cooled CCD camera (Roper Scientific), and a ×40 NA1.3 Plan-Apochromat objective lens (420762-9800, Zeiss). The software used for the acquisition is SlideBook (version 6).

**Immunofluorescence immunohistochemistry (IF IHC) and imaging.** Sections were cut to 4 µm thickness at the Stanford University Histology Service Center and mounted on frosted glass slides. H&E stained sections were obtained from each FFPE block. Deparaffinization, rehydration, and HIER were performed on an ST4020 small linear stainer (Leica). For deparaffinization, slides were baked at 70 °C for 1–1.5 h, followed by rehydration in descending concentrations of ethanol (100% twice, 95% twice, 80%, 70%, $ddH_2O$ twice; each step for 30 s). Washes were performed using a Leica ST4020 Linear Stainer (Leica Biosystems, Wetzlar, Germany) programmed to three dips per wash for 30 s each. HIER was performed in a Lab VisionTM PT module (Thermo Fisher) using Dako Target Retrieval Solution, pH 9 (S236784-2, DAKO Agilent) at 97 °C for 10 min and cooled down to 65 °C. After further cooling to room temperature for 30 min, slides were washed for 10 min three times in Tris-Buffered Saline (TBS), containing 0.1% Tween 20 (Cell Marque; TBS-T). Sections were then blocked in 5% normal donkey serum in TBS-T at room temperature for 1 h, followed by incubation with primary antibodies in the blocking solution. After one overnight incubation of primary antibodies in 4 °C, sections were washed three times with TBS-T and stained with the appropriate secondary antibodies in PBS with 3% bovine serum albumin, 0.4% saponin, and 0.02% sodium azide at room temperature for 1 h. Following this, sections were washed three times with TBS-T and mounted with ProLong Gold Antifade mounting medium with DAPI (Invitrogen). The primary antibodies and final titrations used were rabbit anti-ACE2 (1:100; Abcam ab15348), rabbit anti-ACE2 (1:200; Sigma HPA000288), goat anti-ACE2 (1:100; R&D Systems AF933), rabbit anti-ACE2 (1:100; Abcam ab239924), mouse anti-ACE2 (1:200; R&D Systems MAB933), rabbit anti-ACE2 (1:100; Novus NBP2-67692), mouse anti-acetylated α Tubulin (1:300; Santa Cruz sc-23950), mouse anti-MUC-1 (1:100; NSJ Bio V2372SAF), and rabbit anti-MUC-1 (1:250; Abcam ab109185); mouse anti-MUC5AC (1:200; Abcam ab212636); mouse anti-CD31 (1:300; Novus NBP2-47785); rabbit anti-CD31 (1:50; Abcam ab76533); mouse anti-cytokeratin 8 (1:200; Santa Cruz sc-8020), rabbit IgG isotype control (Abcam ab172730; and same concentration as anti-ACE2 (1:100; Abcam ab15348)). Secondary antibodies include highly cross-adsorbed donkey anti-rabbit Alexa Fluor Plus 647 1:500 (Thermo A32795) and highly cross-adsorbed donkey anti-mouse Alexa Fluor Plus 555 1:500 (Thermo A32773). Fluorescence-immunolabeled images were acquired using a Zeiss AxioImager Z1 microscope or Keyence BZ-X710 fluorescent microscope. Post-imaging processing was performed using FIJI package of ImageJ 1.52t. Figures were organized using Adobe Illustrator.

**In situ hybridization staining.** Tissue sections (4 µm thick) were cut from FFPE tissue blocks and mounted onto glass slides. Slide-tissue sections were baked at 70 °C for 1 h and subsequently soaked in xylene (Thermo Fisher Scientific) for 10 min × 3 rounds. Rehydration and HIER of tissue sections were performed in a similar manner as described above except that rehydrating washes were done for 180 s each. After HIER and cooling, slides were then washed twice with Milli-Q water (Millipore Sigma) subjected to a 10-min protease digestion at 40 °C with Protease III (322337, Bio-Techne) diluted to 1:20 in 1X PBS. Slides were then washed for 2 × 2 min Milli-Q water before a 15-min $H_2O_2$ block at 40 °C (322335, Bio-Techne). Slides were then washed for 2 × 2 min Milli-Q water before an overnight hybridization at 40 °C with probes against the human *ACE2* mRNA (848151, Bio-Techne) or *SARS-CoV-2 Spike* mRNA (848561, Bio-Techne). Amplification of the ISH probes was performed the next day according to manufacturer's protocol (323100, Bio-Techne), with the final deposition of Cyanine 3 for *ACE2* mRNA probe targets (NEL744001KT, Akoya Biosciences). Slides were then processed as described above for IF IHC staining for anti-MUC5AC (Abcam ab212636), ACE2 (Abcam ab15348), and/or cytokeratin 8 (Santa Cruz sc-8020) staining. Fluorescent images were acquired and processed as detailed above.

**Quantification of fluorescence intensity.** Samples within each patient cohort were stained simultaneously with rabbit anti-ACE2 (1:100; Abcam ab15348) and mouse anti-acetylated α-Tubulin (ACTUB 1:300; Santa Cruz sc-23950) using the same master mix and identical incubation times under similar staining conditions described above. Isotype controls were stained with rabbit IgG isotype control (Abcam ab172730) and mouse anti-acetylated α Tubulin (1:300; Santa Cruz sc-23950). Exposure times under fluorescence microscopy were identical for samples

within the same cohort. Quantitation was performed in the FIJI package of ImageJ open source software. Binary masks were created by thresholding the anti-acetylated α-Tubulin channel using selected cutoff values that produce inclusive outlines of the ACTUB staining. Cellular membranes were segmented (outlined) using continuity of high signal areas (area > 1000 pixels) on binary masks as the criteria. The signal within the membrane areas were computed for both the ACE2 channel and for ACTUB channel. The estimates of ACE2 signal were further corrected by subtracting the average membrane signal observed in isotype control from the average ACE2 channel per membrane measurements. For the sake of cross-sample normalization, the ratio of the isotype control-subtracted ACE2 signal divided by the ACTUB signal ("normalized ACE2") for each patient sample was used for further downstream analysis.

**Animal experiments**. Adult C57BL/6J mice (000664, The Jackson Laboratory) were anesthetized by inhalation of 3% isoflurane (Fluriso, Vet One) in 100% oxygen at a delivery rate of 1 L/min using an anesthesia machine (VetEquip). Complete anesthesia was checked by toe pinch reflex, and mice were kept anesthetized throughout the procedure via a face mask connected to an anesthesia machine. The rib cage was cut rostrally from the diaphragm to expose the heart, and an incision was made in the right atrium using scissors. Following insertion of a 27G ½ gauge needle (305109, BD) connected to a 20-ml syringe (302830, BD), mice were transcardially perfused with 20 ml of phosphate-buffered saline (PBS) by slowly pushing the plunger. Mice were then perfused with 1.5 ml/g of 4% (v/v) paraformaldehyde (15710, Electron Microscopy Sciences) in PBS. Trachea tissues were then dissected out and post-fixed in 20 ml of 100% methanol (A412-4, Fisher Scientific) at −20 °C overnight. Fixed tissues were cryoprotected by immersion in graded concentration (10-20-30% (w/v)) of sucrose (S9378, Sigma Aldrich) in PBS until the tissues have sunk to the bottom of each solution. Tissues were then embedded into O.C.T compound (4583, Tissue-Tek), and 10 µm sections were obtained on a cryostat (3050 S, Leica). All mice were maintained under specific pathogen-free conditions at the Stanford animal care facility under 12:12 light-dark cycles at 23 °C with 40% humidity. All animal experiments were conducted in accordance with the institutional AAALAC Guidelines and approved by the Administrative Panel on Laboratory Animal Care (APLAC), Stanford University.

**Statistics and reproducibility**. Analyses were performed with IBM SPSS 23 (IBM Corporation, Armonk, NY) and GraphPad Prism 6.0 (GraphPad Software, La Jolla, CA) software. The two-tailed Student's $t$ test was used for 2-group comparisons. Multiple comparisons for intergroup differences were assessed by Kruskal–Wallis one-way analysis of variance, followed by Dunn's multiple comparison post-hoc test. When integrating data from three institutions, data from each institution were converted to Z-score before applying the above statistical comparison. $\chi^2$ test or Fisher's exact test was used to compare demographics and patients' characteristics between groups. All data are noted as mean ± SD. A $p$-value of <0.05 was considered statistically significant. Images displayed in Figs. 1–4 and Supplementary Fig. 1 represent results from at least two independent experiments.

**Reporting summary**. Further information on research design is available in the Nature Research Reporting Summary linked to this article.

## Data availability
The data sets generated during and/or analyzed during the current study are available from the corresponding author on reasonable request.

## Code availability
The analysis code used to support the findings of this study are available at https://github.com/bmyury/membrane_ACE2_quantitation.

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

## Acknowledgements

We thank members of the Nolan, Nayak, Jackson, Yeh, Tsay, and Stanford Pathology laboratories for helpful discussions and technical assistance. We thank Polly Kavanaugh, Nicole Wang, Carol Valencia, Rebekah Youkhana, Alfred Machicado, Jason Irwin, Camilla Morrison, and Yuka Lee for excellent laboratory and administrative support. We are grateful to faculties of the Stanford Allergy/Immunology Division, especially Drs. Dave Lewis, Yael Gernez, Sean McGhee, and Anne Liu for guidance and accommodations made for Ivan T. Lee's protected research time. We thank Mark Kittisopikul for informal discussion on statistical analysis. We further thank Dr. Ekkehard Hewer, Dr. José Galván, and Sandrine Ruppen (Institute of Pathology, University of Bern, Switzerland) for help with creating tissue microarrays. This work was supported by the Parker Institute for Cancer Immunotherapy (G.P.N.), Food and Drug Administration (HHSF223201610018C and DSTL/AGR00980) (G.P.N.), Fast Grant Funding for COVID-19 Science (G.P.N. and P.K.J.), the Botnar Research Centre for Child Health Emergency Response to COVID-19 Grant (S.J., C.M.S., D.R.M., G.P.N., M.S.M., and A. T.), a Bill and Melinda Gates Foundation COVID-19 Pilot Award (S.J., D.R.M., and G.P. N.), the National Institutes of Health 1R01AI149672-01 (G.P.N.), U54-CA209971 (G.P. N.), P30DK116074 (P.K.J.), the Rachford & Carlotta A. Harris Endowed Chair (G.P.N.), California Institute for Regenerative Medicine (DISC2-09637) (J.V.N.), Defense Advanced Research Project Agency (HR001118S0037-PREPARE-FP-001) (J.V.N.), The Stanford Initiative to Cure Hearing Loss (SICHL) (P.A.G. and J.V.N.), The Operndorf Foundation (P.A.G. and J.V.N.), The PDev Foundation (J.V.N.), Stanford Translational Research and Applied Medicine (TRAM) Pilot Grant (I.T.L.), Thrasher Research Fund Early Career Award (I.T.L.), Stanford Maternal and Child Health Research Institute (MCHRI) Clinical (MD) Trainee Support Award (I.T.L, Ernest and Amelia Gallo Endowed Postdoctoral Fellow), Leukemia & Lymphoma Society Career Development Program (S.J.), and the Swiss National Science Foundation (C.M.S., P400PM_183915).

## Author contributions

I.T.L. conceived and coordinated the study. I.T.L., T.N., C.-T.W., S.J., and P.A.G. designed and performed the experiments. I.T.L., T.N., and C.-T.W. performed the microscopy imaging. I.T.L., T.N., P.A.G., C.-K.L., L.-C.S., C.M.S., D.R.M., P.C., N.A.B., D. Z, S.S.D., A.Y., D.K., K.M.P., R.K., J.B.O., M.A.T., C.H.Y., Y.-T.L., C.-F.L., D.-T.B., G.J.T., Z.M.P., Y-.A.T., M.S.M., A.T., C.-J.T., T.-H.Y., and P.H.H. consented patients, collected, processed, banked, and/or evaluated the human samples. I.T.L., T.N., C.-T.W., Y.G., and S.J. analyzed the data. Y.G. developed and performed computational image processing. Y. G. and T.N. conducted statistical analyses. H.C. provided experimental assistance. T.N. and C.-T.W. prepared the final figures. I.T.L. wrote the manuscript with contributions by T.N., C.-T.W., Y.G., S.J., C.M.S., D.R.M., P.C., G.P.N., J.V.N., and P.K.J. The first co-authors, I.T.L., T.N., C.-T.W., Y.G., and S.J., contributed equally and have the right to list their name first in their CV. Funding and supervision were provided by G.P.N., J.V.N., and P.K.J. All authors reviewed and agreed with the content of this manuscript.

## Competing interests

The authors declare no competing interests.

## Additional information

Ivan T. Lee [1,2,3,17], Tsuguhisa Nakayama[2,4,17], Chien-Ting Wu[5,17], Yury Goltsev[1,17], Sizun Jiang [1,17], Phillip A. Gall[2], Chun-Kang Liao [6], Liang-Chun Shih[7,8,9], Christian M. Schürch [1], David R. McIlwain[1], Pauline Chu[1], Nicole A. Borchard[2], David Zarabanda[2], Sachi S. Dholakia[2], Angela Yang[2], Dayoung Kim[2], Han Chen [1], Tomoharu Kanie[5], Chia-Der Lin[7,8,10], Ming-Hsui Tsai[7,8,10], Katie M. Phillips[2], Raymond Kim[2], Jonathan B. Overdevest [2,11], Matthew A. Tyler[2,12], Carol H. Yan [2,13], Chih-Feng Lin[6], Yi-Tsen Lin[6], Da-Tian Bau[8,9], Gregory J. Tsay[10,14], Zara M. Patel[2], Yung-An Tsou[7,10], Alexandar Tzankov [15], Matthias S. Matter [15], Chih-Jaan Tai[7,10], Te-Huei Yeh[6,16], Peter H. Hwang[2], Garry P. Nolan [1,18✉], Jayakar V. Nayak [2,18] & Peter K. Jackson [1,5,18]

[1]Department of Pathology, Stanford University School of Medicine, Stanford, CA 94305, USA. [2]Department of Otolaryngology–Head and Neck Surgery, Stanford University School of Medicine, Stanford, CA, USA. [3]Division of Allergy, Immunology, and Rheumatology, Department of Pediatrics, Stanford University School of Medicine, Stanford, CA, USA. [4]Department of Otorhinolaryngology, Jikei University School of Medicine, Tokyo, Japan. [5]Baxter Laboratory, Department of Microbiology & Immunology, Stanford University School of Medicine, Stanford, CA, USA. [6]Department of Otolaryngology, National Taiwan University Hospital, Taipei, Taiwan. [7]Department of Otorhinolaryngology, China Medical University Hospital, Taichung, Taiwan. [8]Graduate Institute of Biomedical Sciences, China Medical University, Taichung, Taiwan. [9]Terry Fox Cancer Research Laboratory, Translational Medicine Center, China Medical University Hospital, Taichung, Taiwan. [10]School of Medicine, China Medical University, Taichung, Taiwan. [11]Department of Otolaryngology–Head and Neck Surgery, Columbia University School of Medicine, New York City, NY, USA. [12]Department of Otolaryngology–Head and Neck Surgery, University of Minnesota School of Medicine, Minneapolis, MN, USA. [13]Department of Otolaryngology–Head and Neck Surgery, University of California San Diego School of Medicine, San Diego, CA, USA. [14]Division of Immunology and Rheumatology, Department of Internal Medicine, China Medical University Hospital, Taichung, Taiwan. [15]Pathology, Institute of Medical Genetics and Pathology, University Hospital Basel, University of Basel, Basel, Switzerland. [16]Department of Otolaryngology, College of Medicine, National Taiwan University, Taipei, Taiwan. [17]These authors contributed equally: Ivan T. Lee, Tsuguhisa Nakayama, Chien-Ting Wu, Yury Goltsev, Sizun Jiang. [18]These authors jointly supervised this work: Garry P. Nolan, Jayakar V. Nayak, Peter K. Jackson. ✉email: gnolan@stanford.edu

