## [Peer Review File · Nature Communications]

REVIEWER COMMENTS

Reviewer #1 (Remarks to the Author):

This article described expression and subcellular localization of human ACE2, the SARS-CoV-2 receptor, using immunohistochemistry in archived respiratory samples from human donor. The authors clarified several importance information including: 1) ACE2 is robustly expressed in the motile cilia of the respiratory tract; ACE2 is not expressed in goblet cells of the respiratory tract; no significant differences in ciliary ACE2 expression based on age (≥ 65 years), sex, or smoking status; nasal ciliary ACE2 levels are not increased in patients taking ACEI or ARBs. These results will help to understand how the ongoing SARS-CoV-2 deploy the ACE2 and potential therapeutic strategy for the treatment against the SARS-CoV-2 infection.

The major limitation is there is no analysis in samples infected with SARS-CoV-2.

Reviewer #2 (Remarks to the Author):

I congratulate the authors on a clever, well thought out and comprehensive paper that makes several important discoveries, including:

- the localisation of ACE2 to (molite) cilia;
- the marked expression of ACE2 in nasal tissue
- the importance of not relynmg on RNA-based study methods, which have been misleading.

The translational aspects are important showing that Covid-19 treatmetns sould be targetted to the nose is ground-breaking, but makes perfect sense given their data.

There are limitations with this study and the retrospective approach. However, these are adequately addressed and "forgivable".

While we can always do something better in any study, I do not require any changes to this manuscript.

Peter D Sly

Reviewer #3 (Remarks to the Author):

In the present study, the authors investigated the expression and subcellular localization of angiotensin-converting enzyme 2 (ACE2), the SARS-CoV-2 receptor, within the upper (proximal) and lower (distal) respiratory tracts of human donors using a diverse panel of banked tissues. They discovered that ACE2 expression is located within the motile cilia of airway epithelial cells, which likely represents the initial or early subcellular site of SARS-CoV-2 viral entry during host respiratory transmission. The authors also explored whether ciliary ACE2 expression in the proximal airway depends of demographics data, clinical features, comorbidities, and drug use. They showed the first mechanistic indication that the use of angiotensin-converting enzyme inhibitors (ACEI) or angiotensin II receptor blockers (ARBs) does not increase susceptibility to SARS-CoV-2 infection through enhancing the expression of ciliary ACE2 receptor. They concluded that their results are critical to understand the transmission of SARS-CoV-2 for prevention and control of this virulent pathogen.

Reviewer Comments

1) The major claims of the paper are basically aimed at understanding the mechanisms of injury induced by CoV-2 in the proximal and distal airways. The design of the work is coherent to achieve the objectives and uses an appropriate methodology aimed at obtaining results consistent with the objectives. The figures used to demonstrate the results are extremely illustrative, of good quality and total control of the immunohistochemistry and immunofluorescence standardization. Figure captions are self-explanatory and allow the reader to clearly view the results.

2) The investigation is novel and certainly will be of interest to others in the community and the wider field not only in COVID-19 pandemic but also for patients with immobile cilia syndrome. Regarding the immobile cilia syndrome, it would have been very useful if the authors had complemented their study with Transmission Electron Microscopy, generally used to diagnose the syndrome, and thereby document submicroscopic changes induced by CoV-2 that would justify the cellular alterations seen by immunohistochemistry and immunofluorescence.

3) The author's conclusions are original and supported by the results obtained.

4) The documentation presented by the authors about the results is convincing and represents, in my view, "the heart of the paper", thus reinforcing the conclusions.

5) The statistical analysis is valid and clearly appropriated to obtain the results. The paper reflects the ability of a researcher to reproduce the work, given the level of detail provided.

6) In summary, according to my point of view as a Lung Pathologist the paper will influence the research field in which it is inserted.

7) As mentioned earlier, my only concern with the paper was not using TEM for submicroscopic documentation of possible ciliary changes.

Response to referees

REVIEWER COMMENTS

Reviewer #1 (Remarks to the Author):

This article described expression and subcellular localization of human ACE2, the SARS-CoV-2 receptor, using immunohistochemistry in archived respiratory samples from human donor. The authors clarified several importance information including: 1) ACE2 is robustly expressed in the motile cilia of the respiratory tract; ACE2 is not expressed in goblet cells of the respiratory tract; no significant differences in ciliary ACE2 expression based on age (≥ 65 years), sex, or smoking status; nasal ciliary ACE2 levels are not increased in patients taking ACEI or ARBs. These results will help to understand how the ongoing SARS-CoV-2 deploy the ACE2 and potential therapeutic strategy for the treatment against the SARS-CoV-2 infection.

The major limitation is there is no analysis in samples infected with SARS-CoV-2.

Response: We agree with Reviewer 1 that adding samples infected with SARS-CoV-2 would provide additional context to our findings. We have successfully obtained sinonasal tissue infected with SARS-CoV-2 and provide representative images in Figure 3 demonstrating the presence of the SARS-CoV-2 Spike mRNA within ciliated epithelial cells in relation to ACE2-positive motile cilia. Given the motile cilia densely cover the outer apical surface of ciliated epithelial cells, this finding further suggests that the SARS-CoV-2 virus enters ciliated epithelial cells by first binding ACE2 located in the motile cilia.

Reviewer #2 (Remarks to the Author):

I congratulate the authors on a clever, well thought out and comprehensive paper that makes several important discoveries, including:

- the localization of ACE2 to (motile) cilia;
- the marked expression of ACE2 in nasal tissue;
- the importance of not relying on RNA-based study methods, which have been misleading.

The translational aspects are important showing that Covid-19 treatments should be targeted to the nose is ground-breaking, but makes perfect sense given their data.

There are limitations with this study and the retrospective approach. However, these are adequately addressed and "forgivable".

While we can always do something better in any study, I do not require any changes to this manuscript.

Peter D Sly

Response: We appreciate Dr. Sly's kind comments and nice summary of our findings.

Reviewer #3 (Remarks to the Author):

In the present study, the authors investigated the expression and subcellular localization of angiotensin-converting enzyme 2 (ACE2), the SARS-CoV-2 receptor, within the upper (proximal) and lower (distal) respiratory tracts of human donors using a diverse panel of banked tissues. They discovered that ACE2 expression is located within the motile cilia of airway epithelial cells, which likely represents the initial or early subcellular site of SARS-CoV-2 viral entry during host respiratory transmission. The authors also explored whether ciliary ACE2 expression in the proximal airway depends of demographics data, clinical features, comorbidities, and drug use. They showed the first mechanistic indication that the use of angiotensin-converting enzyme inhibitors (ACEI) or angiotensin II receptor blockers (ARBs) does not increase susceptibility to SARS-CoV-2 infection through enhancing the expression of ciliary ACE2 receptor. They concluded that their results are critical to understand the transmission of SARS-CoV-2 for prevention and control of this virulent pathogen.

Reviewer Comments

- 1) The major claims of the paper are basically aimed at understanding the mechanisms of injury induced by CoV-2 in the proximal and distal airways. The design of the work is coherent to achieve the objectives and uses an appropriate methodology aimed at obtaining results consistent with the objectives. The figures used to demonstrate the results are extremely illustrative, of good quality and total control of the immunohistochemistry and immunofluorescence standardization. Figure captions are self-explanatory and allow the reader to clearly view the results.
- 2) The investigation is novel and certainly will be of interest to others in the community and the wider field not only in COVID-19 pandemic but also for patients with immobile cilia syndrome. Regarding the immobile cilia syndrome, it would have been very useful if the authors had complemented their study with Transmission Electron Microscopy, generally used to diagnose the syndrome, and thereby document submicroscopic changes induced by CoV-2 that would justify the cellular alterations seen by immunohistochemistry and immunofluorescence.

Response: We appreciate Reviewer 3's generous compliments of our work and expertise in airway pathology. Although immobile cilia syndrome is not a topic of our current manuscript, we absolutely agree with Reviewer 3 that it would be interesting to explore the biological relevance of our findings in relation to patients with immobile cilia syndrome in future studies if such rare samples can be acquired.

- 3) The author's conclusions are original and supported by the results obtained.
- 4) The documentation presented by the authors about the results is convincing and represents, in my view, "the heart of the paper", thus reinforcing the conclusions.
- 5) The statistical analysis is valid and clearly appropriated to obtain the results. The paper reflects the ability of a researcher to reproduce the work, given the level of detail provided.
- 6) In summary, according to my point of view as a Lung Pathologist the paper will influence the research field in which it is inserted.
- 7) As mentioned earlier, my only concern with the paper was not using TEM for submicroscopic documentation of possible ciliary changes.

Response: We appreciate the reviewer's suggestion on using TEM if we can obtain samples from patients with immobile cilia syndrome in future studies.

Other minor revisions that we have made to this manuscript:

- We added sentences to describe the addition of SARS-CoV-2-infected tissue in Lines 93-94, 146-155, 301-304, and 357-360.
- We added additional authors who contributed to experiments involving SARS-CoV-2 infected upper airway tissue.
- We deleted the reference, Mehra et al. NEJM 2020, from our discussion section and edited Lines 296-300 to reflect this deletion, as Mehra et al. NEJM 2020 has since been retracted. The contents of this discussion section remain unchanged as our findings that ACE2 is not increased by ACEI/ARBs continue to be supported by Reynolds et al. NEJM 2020, Mancina et al. NEJM 2020, and 2 studies that were published after our initial manuscript submission, Fosbøl et al. JAMA 2020 and de Abajo et al. Lancet 2020. We added references to these 2 latter papers.
- We updated our funding sources.

REVIEWERS' COMMENTS:

Reviewer #1 (Remarks to the Author):

The authors have answered my question and provided a new figure using COVID-19 patient autopsy tissues. I've no further question.

Zhengli Shi